# Twenty-Five Years of Contemplating Genotype-Based Hereditary Hemochromatosis Population Screening

**DOI:** 10.3390/genes13091622

**Published:** 2022-09-09

**Authors:** Jörg Schmidtke

**Affiliations:** 1Hannover Medical School, Institute of Human Genetics, Carl-Neuberg-Strasse 1, D-30623 Hannover, Germany; schmidtke.joerg@mh-hannover.de; 2Amedes MVZ Wagnerstibbe, Georgstrasse 50, D-30159 Hannover, Germany

**Keywords:** hereditary hemochromatosis, population screening, genotype-based screening, phenotypic screening

## Abstract

Hereditary hemochromatosis (HH) is a rather frequent, preventable disease because the progressive iron overload affecting many organs can be effectively reduced by phlebotomy. Even before the discovery of the major gene, *HFE*, in 1996, hemochromatosis was seen as a candidate for population-wide screening programmes. A US Centers of Disease Control and the National Human Genome Research Institute expert panel convened in 1997 to consider genotype-based HH population-wide screening and decided that the scientific evidence available at that time was insufficient and advised against. In spite of a large number of studies performed within the last 25 years, addressing all aspects of HH natural history, health economics, and social acceptability, no professional body worldwide has reverted this decision, and HH remains a life-threatening condition that often goes undetected at a curable stage.

## 1. Introduction

Hereditary hemochromatosis (HH (HFE1, MIM# 235200)) is an autosomal recessive disorder characterized by increased iron absorption and iron overload affecting liver, heart, pancreas, joints, and other organs, with males far more strongly affected than females. Cardiovascular manifestations and severe liver disease are the leading causes of morbidity and mortality in hemochromatosis patients [1,2]. Cardiac hemochromatosis is characterized by arrhythmias, dilated cardiomyopathy, reduced ejection fraction, and reduced fractional shortening. Deposition of iron may occur in the entire cardiac conduction system, especially the atrioventricular node. Cardiac hemochromatosis should be considered in any patient with unexplained heart failure [3]. Early diagnosis is desirable, because iron removal by serial phlebotomy is highly effective, safe, and inexpensive, leading to a normal life expectancy when started before the development of clinical complications [4,5,6].

Around 80% to 90% of HH cases [7,8,9] are attributable to homozygosity for a G to A transition at nucleotide 845 of the *HFE* gene, resulting in a cysteine to tyrosine substitution at amino acid 282 (C282Y) [7]. (The genetic heterogeneity of hemochromatosis is discussed by Le Gac and Férec [10].) The frequency of *HFE* C282Y homozygosity among non-Hispanic whites is estimated to be about 1 in 300 [11], up to 1 in 150 in people of Northern European descent [12], and at the highest at 1 in 83 in Ireland [13]. Because of its high frequency, the availability of simple test methods, and effective prevention and treatment, HH has for long been considered as a candidate for the implementation of population or targeted screening, either based on biochemical serum iron markers or based on genotype [14,15,16,17].

However, in view of existing uncertainties regarding the prevalence and penetrance of *HFE* mutations, and the optimal care of asymptomatic people carrying pathogenic *HFE* variants, as first proposed by Burke et al. [18] on the basis of a US Centers of Disease Control (CDC) and Prevention and the National Human Genome Research Institute (NHGRI) expert meeting gathered in 1997, a need for further research was generally recognized in order to study the genotype–phenotype correlations in HH and to investigate the ethical, social, and psychological effects of genotype-based testing before population-based screening programmes were eventually established. Thus, a large series of population-based studies was conducted world-wide, variably emphasizing the biochemical and clinical penetrance of HH-associated genotypes, health economic aspects, and societal acceptability.

This review does not provide a systematic and comprehensive account of the vast literature on this topic. It is intended to illustrate the controversial discussion surrounding population screening for HH with some emphasis on societal response and professional recommendations using selected examples from the literature.

## 2. Biochemical and Clinical Penetrance

Elevated levels of serum ferritin and transferrin saturation (“biochemical penetrance”) were found to occur in 40 to 60% of female homozygotes and in 75 to 100% of male homozygotes [16,19,20,21,22]. An important hallmark was the finding, in a large prospective study conducted in Australia, that serious iron overload as defined by serum ferritin levels of 1000 ug/L and above occurs in 35% of male and 6% of female homozygotes at a median age of 65 years [23].

The question remained as to what proportion of homozygotes would become clinically affected by HH symptoms (“clinical penetrance”). This question was addressed in highly divergent manners. Beutler et al. [20] assessed the existence of the full-blown clinical picture of hemochromatosis, namely, the combination of liver disease, heart failure, diabetes mellitus, and bronze skin in a cohort of 152 HFE homozygotes identified in a large screening study conducted in the USA comprising 41,038 individuals attending a health appraisal clinic. The authors came to the conclusion that with less than 1% of homozygotes developing “frank clinical hemochromatosis”, clinical penetrance was much lower than generally thought. Whitlock et al. [24] used the cross-sectional prevalence of liver disease as a proxy of clinical penetrance and found that only 1.4% of newly diagnosed homozygotes had biopsy-confirmed liver cirrhosis. The “rarity” of clinical penetrance presumed from these and other studies prompted the U.S. Preventive Services Task Force [25] to advise against general population screening for hemochromatosis, a recommendation unaltered up to the present day (see below).

Grosse et al. [26] reviewed the epidemiologic evidence from both prospective and cross-sectional population-based studies on the clinical prevalence of HFE homozygosity in terms of the cumulative risk for severe liver disease. They pointed out that cross-sectional prevalence studies can be misleading, and that the suitable epidemiologic measure of clinical penetrance should be life-time incidence (as, for example in colorectal cancer). They concluded on the basis of published data “that roughly 1 in 10 male *HFE* C282Y homozygotes is likely to develop severe liver disease during his lifetime unless iron overload is detected early and treated”, a value much higher than the estimates previously quoted in the scientific literature. They pointed out that the overall clinical penetrance in terms of iron-overload-related symptoms is around 28% in males. They acknowledged that population screening for *HFE* C282Y homozygosity faces multiple barriers and suggested, as a potentially effective strategy for increasing the early detection and prevention of clinical iron overload and severe disease, “to include *HFE* C282Y homozygosity in lists of medically actionable gene variants when reporting the results of genome or exome sequencing”.

In any case, given that a substantial proportion of HH homozygotes apparently faces life-threatening disease if untreated, it is puzzling that cohorts of elderly people are not depleted of this genotype, implying that, overall, HH homozygotes do not have shorter life expectancy [27,28,29,30,31]. One possible explanation for this apparent paradox would be that in the absence of relevant organ manifestation, HH homozygosity increases life expectancy by mechanisms such as a genotypic association with reduced LDL cholesterol [31,32,33].

## 3. Health Economic Aspects

Health economic studies modelling cost-effectiveness of population screening programmes produced seemingly contradictory results. 

In an early study by Phatak et al. [34], a model was constructed to compare the costs and outcomes of a strategy of performing biochemical tests on cohorts of 30-year-old men with that of awaiting symptomatic disease. They found a favourable cost/effectiveness ratio over a wide range of assumptions.

Rogowski [35] addressed cost/effectiveness of targeted (male offspring of male homozygotes) and population-wide screening, with both phenotypic (biochemical) and genotypic methodologies, and from the perspective of the German health care system. He found that while reducing preventable deaths, a population-wide programme would amount to a cost/effectivity ratio of between 124,000 (transferrin saturation testing and genotyping, if TS is elevated) and 161,000 EUR/LYG (genotype screening), which does not meet frequently cited benchmarks of cost/effectiveness. It is the only study reviewed by de Graaff et al. [36] with such a negative result. However, as pointed out by Grosse et al. [26], using data from cohort studies rather than from cross-sectional studies like those in the Rogowski study, the clinical penetrance of severe liver disease among older males and thus the potential effects may be 2.5–3 times higher than estimated by Rogowski [35]. In addition, de Graaff et al. [36] have argued that in the Rogowski model, screening costs may have been overestimated, as Rogowski assumed annual serum ferritin monitoring, which is not a universal strategy in accordance with the EASL guidelines. In the absence of further studies, Rogowski [35] neither included evidence on current practice of regular testing for C272Y homozygotes in Germany, nor evidence on potentially unnecessary additional diagnostic procedures. Thus, the model by Rogowski [35] may need an update [37].

De Graaff et al. [36] systematically reviewed 38 health economic studies conducted for HH screening programmes to that date. They found that whilst most studies concluded screening was cost effective compared with no screening, methodological flaws would limit the quality of these findings. Assumptions regarding clinical penetrance, effectiveness of screening, health-state utility values, exclusion of early symptomatology, and quantification of costs associated with HH were identified as key limitations. Treatment studies showed that therapeutic phlebotomy was the most cost-effective intervention. The authors concluded that there is a paucity of high-quality health economic studies relating to HH, and that the development of a comprehensive HH cost/effectiveness model utilising health-state utility values is required to determine whether screening is worthwhile.

The most recently published cost/effectiveness model of population screening strategies for HH supports the notion that routine genotyping of Australian males of European descent at 30 years of age is highly cost effective [38].

## 4. Societal Acceptability

Hicken et al. [39] investigated the attitudes around testing for hemochromatosis in 118 young adults and 50 older adults. Participants were informed about hemochromatosis, the transferrin saturation measurement, and *HFE* genotyping. They found that over 80% of participants would be willing to undergo either test if offered. Biochemical testing was preferred by the majority because it would provide information about current health. Young adults “were more likely to report disadvantages of genetic testing and were more concerned about potential negative psychological effects”, and old adults “were more concerned about potential discrimination”. Altogether, young and old adults viewed genetic testing as beneficial, and HFE testing would be accepted if offered as part of a screening program.

Hicken et al. [40] examined attitudes regarding genetic testing and psychosocial outcomes of *HFE* genotyping for hemochromatosis; 87 persons with hemochromatosis (patients), who underwent *HFE* genotyping, and 50 controls, who had not undergone *HFE* genotyping, reported attitudes about benefits and disadvantages of genetic testing and their understanding of genetics and hemochromatosis. Most participants felt genetic testing to be beneficial and described few negative aspects of testing. Controls expected more anxiety, depression, and anger related to a positive genetic test than patients. Most patients were compliant with the treatment regimen. The authors concluded that *HFE* genotyping appears to be viewed positively and would be generally accepted were it offered as part of a screening program for hemochromatosis. Persons who have not undergone genetic testing may overestimate their emotional responses to a positive test result. In the present hemochromatosis patients, few reported that *HFE* genotyping was accompanied by negative psychosocial outcomes.

Delatycki et al. [22] assessed, in view of the controversies surrounding genetic screening for HH, whether such screening was suitable for communities. Screening for the Cys282Tyr *HFE* mutation was offered to individuals at their workplace, and 11,307 individuals were screened. No increase in anxiety was recorded among homozygotes for the Cys282Tyr mutation or non-homozygotes. Nearly all homozygous individuals identified (46 of 47) took steps to treat or prevent iron accumulation. They concluded that population genetic screening for HH can be practicable and acceptable.

Gason et al. [41] sought to establish attitudes towards genetic susceptibility screening in Australian secondary schools, with hereditary hemochromatosis as the model condition. Attitudes towards genetic screening in schools and knowledge of genetic and clinical features of hemochromatosis, as well as the likelihood of accepting a genetic susceptibility test for hemochromatosis, were measured in students (*n* = 748), parents (*n* = 179), and staff (*n* = 89). Participants felt positive about genetic screening for disease susceptibility in schools. Knowledge following education was high with no significant differences between participants of each group, and 68% of students would take the test if offered. The authors pointed out that genetic susceptibility screening in schools is a novel concept and expected, on the basis of their study, that it could be a public health success with the support of the community.

Stuhrmann et al. [42] conducted a pilot study on DNA-based population screening of hereditary hemochromatosis (HH) in Germany. A collaborating health insurance organization informed their members about the possibility to participate voluntarily. A total of 5882 members expressed their interest and received information on clinical and genetic aspects of HH and the aim of the project; 3961 of these requested *HFE* genotyping. After genotype results had been communicated via the participants’ general practitioner, in order to assess the psychosocial impact of *HFE* genotyping, self-administered questionnaires were sent to all homozygous (*n* = 67) and heterozygous (*n* = 485) as well as 448 wild-type study participants. Additionally, questionnaires were sent to 8000 randomly selected members of the insurance in order to investigate their attitude toward genetic testing. 631 (63.1%) of the test participants and 2141 (26.8%) of the randomly chosen members responded; 59.1% of the members would accept predictive genetic testing, and 3.7% objected to such tests in principle, while 69.9% of the tested individuals thought that participation in the pilot study was probably beneficial for them, and only 1% thought that it was probably harmful. In total, 94.6% of the participants judged their decision to have participated in the pilot study as right, and only 0.3% as probably wrong. These figures may be skewed by the fact that 42.6% of the homozygotes had prior knowledge of their HH clinical status. It was concluded in this study that genotype-based screening for HH is generally accepted and was perceived as beneficial, with negative psychosocial consequences being rare and presumably preventable by delivering appropriate pretest and posttest information.

In view of the existing uncertainties regarding clinical penetrance this survey included a question addressing this point. It was phrased in a more general way: What level of probability of manifestation of a serious disease should there be in order for a predictive test to be covered by the insurance? In total, 40% of the responders thought that the level of probability should be at least 10% [42].

Power et al. [43] investigated the differences in the psychological effects of genetic screening for hemochromatosis (*HFE* mutation analysis) in 2654 participants of the large “Hemochromatosis and Iron Overload Screening (HEIRS) Study”, conducted in Canada and the USA, with altogether 101,168 participants. Regardless of mutation result or country, participants reported similar changes in general and mental health. Over 50% of C282Y homozygote participants from both countries experienced worry in response to testing. “Thus, although not serious enough to affect individuals’ mental or physical health, there was evidence of at least one element of negative emotional response to the genetic testing”.

Another large study by Delatycki et al. [44] assessed the acceptability and feasibility of genetic screening for hereditary hemochromatosis in high-school students. Screening was offered to 17,638 students. Outcomes assessed through questionnaires were uptake of screening; change in scores of validated anxiety, affect, and health perception scales over time; knowledge; and iron indices in C282Y homozygous individuals. A total of 5757 (32.6%) students underwent screening, and 28 C282Y-homozygous individuals (1 in 206) were identified. “There was no significant change in measures of anxiety, affect or health perception in C282Y homozygous or non-homozygous individuals. … Genetic population-based screening for a preventable disease can be offered in schools in a way that results in minimal morbidity for those identified at high risk of disease. The results of this study are not only relevant for hemochromatosis, but for other genetic markers of preventable disease such as those for cardiovascular disease and cancer”.

## 5. Professional Recommendations

To date, no population-based hemochromatosis screening programmes have been established. The following relevant national and international professional organisations have stated their views on this topic.

### 5.1. U.S. Preventive Services Task Force (USPTF)

The U.S. Preventive Services Task Force (USPTF) is an independent, volunteer panel of national experts in disease prevention and evidence-based medicine. The Task Force works to improve the health of people nationwide by making evidence-based recommendations about clinical preventive services. Regarding HH screening, a statement was issued in 2006 [25] and has not been revised since then. It recommends against routine genetic screening for hereditary hemochromatosis in the asymptomatic general population. The USPSTF found fair evidence, at the time of issuing that statement, that a low proportion of individuals with a high-risk genotype (C282Y homozygote at the *HFE* locus, a mutation common among white populations presenting with clinical symptoms) manifest the disease. There was poor evidence that early therapeutic phlebotomy improves morbidity and mortality in screening-detected versus clinically detected individuals. It was thought that screening could lead to identification of a large number of individuals who possess the high-risk genotype but may never manifest the clinical disease. This may result in unnecessary surveillance, labelling, unnecessary invasive work-up, anxiety, and, potentially, unnecessary treatments. The USPSTF concluded that the potential harms of genetic screening for hereditary hemochromatosis outweigh the potential benefits.

### 5.2. American Association for the Study of Liver Disease (AASLD)

The American Association for the Study of Liver Disease (AASLD) is a leading organization of scientists and health care professionals committed to preventing and curing liver disease. AASLD does not recommend average risk population screening for HH. It does recommend screening (iron studies and HFE mutation analysis) of first-degree relatives of patients with HFE-related HH to detect early disease and prevent complications [45].

### 5.3. European Association for the Study of the Liver (EASL)

The purpose of the association is to promote communication between European professionals interested in the liver and its disorders. In their clinical practice guideline issued in 2010, they stated: “Genetic screening for HFE–HC is not recommended, because disease penetrance is low and only in few C282Y homozygotes will iron overload progress” [46]. With reference to studies published at that time, the guideline assumed a 10–33% chance of C282Y homozygotes to eventually develop hemochromatosis-associated morbidity. This practice guideline was recently revised [47]. With reference to a letter to the editor of the *New England Journal of Medicine* by Waalen and Beutler [48] and the ensuing discussion it erroneously states: “Haemochromatosis penetrance leading to endorgan damage in patients with p.C282Y homozygosity is only 50% in population-based studies”. It continues: “This represents significant limitations for genotypic screening for hemochromatosis”. While family screening is advocated, population screening is not recommended. 

### 5.4. US Centers for Disease Control and Prevention

The US Centers for Disease Control and Prevention (CDC) quote the U.S. Preventive Services Task Force recommending against routine genetic screening for hereditary hemochromatosis in the asymptomatic general population, but stating that individuals with a family member, especially a sibling, who is known to have hereditary hemochromatosis should be counselled regarding genetic testing [49].

### 5.5. The American College of Medical Genetics and Genomics

The American College of Medical Genetics and Genomics (ACMG) is the US interdisciplinary professional membership organization that represents the interests of the entire medical genetics team including clinical geneticists, clinical laboratory geneticists, and genetic counsellors. Starting in 2013, and as of now with annual updates, ACMG publishes guidance for reporting secondary findings in the context of clinical exome and genome sequencing in the sense of an “opportunistic screening” for “actionable” pathogenic gene variants [50]. Actionability is given when established interventions exist aiming at preventing or significantly reducing morbidity and mortality. Following a suggestion of Grosse et al. [26] (see above), their actionable genes list now contains the *HFE* gene. As Laberge [51] has pointed out, from a public health perspective, such “opportunistic screening” is “not the most efficient approach” to reduce HH morbidity, as currently only a small minority of individuals receive exome or genome sequencing in a clinical setting. The ACMG concept of offering screening for actionable genes only to those who underwent exome or genome sequencing for other reasons but not recommending the actionable gene panel as a test offered to the general population has been criticized in general [52].

### 5.6. Others

None of the following organisations have issued any specific statement on the topic of population screening for HH: the international research network Cochrane, the European Society of Human Genetics (ESHG), The UK National Institute for Health and Care Excellence (NICE), and the World Health Organization (WHO).

## 6. To Screen, and How, or Not to Screen?

Decisions to introduce population-wide screening programmes must be based on solid evidence regarding effectiveness and a positive balance of benefits and harms [53]. While the general WHO screening criteria [54] were generally seen as fulfilled soon after the discovery of the gene mutations causing HH, evidence regarding the natural history of the disease, agreement on whom to test, costs, positive predictive value, and social acceptability was seen to be missing or incomplete [18,51]. Given that none of the professional organisations cited above has so far reviewed their opinion in favour of population screening, it must be concluded that evidence in favour of screening remains insufficient. It can be assumed that the allegedly low clinical penetrance of the HH-associated genotypes (equalling a low predictive value of a genotype-based screening regimen), the ensuing dangers of overdiagnosis and overtreatment, as well as the overall low efficiency of genotype-based testing [55] (see below) remain the major obstacles. Apparently, the work of Grosse et al. [26] four years ago has not put an end to the debate of which HH-associated symptoms and what levels of penetrance would warrant a positive decision towards population screening, even though there is evidence that a substantial proportion of people would opt for genotype-based screening if penetrance was in the order of 10% [42]. The inclusion of HH-related genotypes in the “opportunistic screening” advocated by ACMG [50] is seemingly “a consolation prize for a condition once thought to be a prime candidate for population-based screening” [51]. 

Most professional organizations (see above) do, however, advocate targeted phenotype- or genotype-based screening for HH, in particular in first-degree family members of patients with HH (“cascade screening”). Given that the clinical penetrance of HH-related genotypes is long known to be highly variable also within families [56], such recommendations seem to collide with the notion that this feature argues against population screening. The efficiency and efficacy of cascade vs. population screening have been discussed by Krawczak et al. [55]. Given that the carrier risk of close relatives of known carriers is generally higher than the population risk, cascade screening must be more efficient than population screening in the sense that fewer individuals have to be genotyped per the detected carrier. The efficacy of cascade screening, as measured by the overall proportion of carriers detected in a given population, is, however, lower than that of population-wide screening. The authors showed that high detection rates are achievable, however, when screening is performed to detect covert homozygotes for frequent recessive mutations with reduced penetrance. For HH, up to 40% of at-risk individuals may be identifiable through screening of first- to third-degree relatives of overt carriers (i.e., patients); the efficiency of this screening strategy was found to be approximately 50 times higher than that of population-wide screening.

Another way of enhancing the efficiency of genotypic HH screening by focusing on individuals with elevated prior risk could be to use biochemical measures of iron overload as the frontline approach [51]. Some health care systems, e.g., France [57], Germany [58], the UK [59], and Australia [60], offer, for all members of their obligatory health insurance systems, or citizens, respectively, regular health check-ups, which include blood tests for diabetes and hypercholesterolemia. It would be a relatively easy task to organize effective HH-related biochemical tests as an add-on to these check-ups, especially when offered to young adults. Sequential genotyping of individuals above set threshold values would then enable differentiation of true HH cases from people with iron-overload for other reasons, including non *HFE*-related types of hemochromatosis [10], which are particularly relevant for possible cardiac involvement [47]. A favourable cost-effectiveness ratio of such a sequential approach was already demonstrated by Rogowski [35] and is likely to yet become more favourable [37] in the light of the arguments by Grosse et al. [26] and de Graaff et al. [38].

With all experts in agreement about HH being a frequent disease, completely curable at low cost when detected early enough, we should not wait another 25 years to implement suitable measures. There are too many patients with this condition who have missed their opportunity for leading a healthy life due to a lack of awareness by their doctors and missing collective actions [61].

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
