# Peer review of "Twenty-Five Years of Contemplating Genotype-Based Hereditary Hemochromatosis Population Screening"

_genes, 2022, doi:10.3390/genes13091622_

Round 1

Reviewer 1 Report

The paper can be treated as an important contribution to the discussion on the type and scale of screening tests in hereditary haemochromatosis. However, data relevant to the assertion that “HH remains to be a life-threatening condition that often goes undetected at a curable stage” (as authors conclude) is  clearly missing.

Please add the source(s) of this statement relevant to the published results.

Author Response

I have added a reference [61] to HH underdiagnosis at the very end of the manuscript.

Reviewer 2 Report

The topic is important, up-to-date, and still rise a discussion. The defined by the author role of the manuscript, which is to  "illustrate the controversial discussion surrounding 60 population screening for HH" is fulfilled and several aspects of the topic are presented. The conclusion are resulting from the text.

I have just a few minor comments:

Line 26 Please provide the MIM number in the correct format: MIM235200

Line 39 Between around-either between or around

Line 167-178,197-203, 215-222- I would shorten and rephrase the citations

Line 343-344  none of the above professional organizations cited above-"above" is repeated

Line 398  the name Krawczak is misspelled

Author Response

The MIM# Format has been corrected. Between/around was replaced by around ... to The three citations were paraphrased and shortened One "above" was deleted "Krawczak" is now spelled correctly